# Diverse and Synergistic Actions of Phytochemicals in a Plant-Based Multivitamin/Mineral Supplement against Oxidative Stress and Inflammation in Healthy Individuals: A Systems Biology Approach Based on a Randomized Clinical Trial

**DOI:** 10.3390/antiox13010036

**Published:** 2023-12-23

**Authors:** Seunghee Kang, Youjin Kim, Yeonkyung Lee, Oran Kwon

**Affiliations:** 1Department of Nutritional Science and Food Management, Ewha Womans University, Seoul 03760, Republic of Korea; nutrishee@gmail.com; 2Logme Inc., Seoul 03182, Republic of Korea; 3Innovation and Science, Amway Korea Ltd., Seoul 06414, Republic of Korea; 4Department of Nutritional Science and Food Management, Graduate Program in System Health Science and Engineering, Ewha Womans University, Seoul 03760, Republic of Korea

**Keywords:** phytochemical, network analysis, biological process, human clinical trial, oxidative stress, inflammation

## Abstract

Traditional clinical methodologies often fall short of revealing the complex interplay of multiple components and targets within the human body. This study was designed to explore the complex and synergistic effects of phytochemicals in a plant-based multivitamin/mineral supplement (PBS) on oxidative stress and inflammation in healthy individuals. Utilizing a systems biology framework, we integrated clinical with multi-omics analyses, including UPLC-Q-TOF-MS for 33 phytochemicals, qPCR for 42 differential transcripts, and GC-TOF-MS for 17 differential metabolites. A Gene Ontology analysis facilitated the identification of 367 biological processes linked to oxidative stress and inflammation. As a result, a comprehensive network was constructed consisting of 255 nodes and 1579 edges, featuring 10 phytochemicals, 26 targets, and 218 biological processes. Quercetin was identified as having the broadest target spectrum, succeeded by ellagic acid, hesperidin, chlorogenic acid, and quercitrin. Moreover, several phytochemicals were associated with key genes such as *HMOX1*, *TNF*, *NFE2L2*, *CXCL8*, and *IL6*, which play roles in the Toll-like receptor, NF-kappa B, adipocytokine, and C-type lectin receptor signaling pathways. This clinical data-driven network system approach has significantly advanced our comprehension of a PBS’s effects by pinpointing pivotal phytochemicals and delineating their synergistic actions, thus illuminating potential molecular mechanisms.

## 1. Introduction

Since 2010, more than 18,000 clinical studies have probed the role of food or diet in bolstering health and curbing chronic illnesses [1]. Yet, they have not adequately unraveled the intricate interplay of food components with their biological targets and the associated molecular dynamics. Consequently, there is an increasing dialogue advocating for a transition in clinical research from the reductionist method to a more comprehensive, holistic approach [2]. Reductionism breaks down complex systems into basic elements, assuming that a system’s full scope can be understood by its individual parts [3]. However, contemporary research supports Aristotle’s assertion that “the whole is greater than the sum of its parts,” with the reductionist model often failing to illuminate the complex interplay between dietary components and the ensuing cellular and molecular responses within living systems. In contrast, a holistic perspective, aiming to understand the intricate network of relationships between nutrition and health, serves to enhance and broaden the traditional reductionist viewpoint [4].

This shift towards a more integrative perspective is increasingly apparent in clinical research. Traditional studies have narrowly focused on specific biomarkers, often overlooking the intricate interactions between the numerous components in foods or diets and the diverse targets within human physiology. Our prior research, for instance, examined the efficacy of a plant-based multivitamin/mineral supplement (PBS) in reducing plasma reactive oxygen species (ROS) levels and lessening DNA damage in adults with a low consumption of fruits and vegetables [5]. Characterized by its rich spectrum of phytochemicals—plant-derived secondary metabolites posited to extend health benefits beyond essential nutrition—a PBS represents a complex nutraceutical entity. However, the current scope of clinical research methodologies falls short of fully demystifying the complex interplay and potential synergistic effects exerted by these constituents on human physiology, particularly in augmenting the supplement’s antioxidant and anti-inflammatory responses. Consequently, there is a compelling need for a more sophisticated research paradigm to unravel the nuanced mechanisms through which PBS components individually target specific physiological pathways. This urges the integration of computational biology and multi-omics strategies within conventional clinical research infrastructures to yield a more holistic understanding.

The purpose of this study was dual: firstly, to integrate clinical data from PBS supplementation with multi-omics information, thereby enabling advanced in silico network analyses; and secondly, to elucidate the fundamental mechanisms and synergistic effects among the various phytochemicals in the PBS and their interactions with a range of physiological targets. In pursuit of this, we scrutinized changes in the endogenous metabolite configurations in plasma and transcriptomic alterations in peripheral blood mononuclear cells (PBMCs) before and after PBS administration, highlighting the modified parameters related to oxidative stress and inflammation. Subsequently, we constructed an association network analysis employing the context-oriented directed association (CODA) software, version 3.0 (Bio-Synergy Research Center, Daejeon, Republic of Korea), a specialized tool designed to integrate context into biological linkages using established scientific literature [6]. This approach significantly enhanced our ability to deepen and strengthen traditional research by identifying the harmonious interactions between the PBS phytochemicals and various molecular targets in combating oxidative and inflammatory stresses. Finally, we explored the signaling pathways associated with oxidative stress and inflammation to gain a deeper understanding of the underlying mechanisms.

## 2. Materials and Methods

### 2.1. Test Materials

The specifics of the test materials and dosage details are documented in a previous publication [5]. In brief, the daily dose of the PBS comprised 12 tablets containing an array of 14 vitamins (700 μg retinol equivalents of A, 2.4 mg of B1, 2.8 mg of B2, 3 mg of B6, 4.8 μg of B12, 200 mg of C, 10 μg of D, 22 mg α-tocopherol equivalents of E, 55 μg of K, 3 mg of β-carotene, 60 μg of biotin, 500 μg of folate, 30 mg of niacin, and 10 mg of pantothenic acid) and 10 minerals (700 mg of calcium, 50 μg of chromium, 0.4 mg of copper, 75 μg of iodine, 6 mg of iron, 3 mg of manganese, 220 mg of magnesium, 25 μg of molybdenum, 55 μg of selenium, and 12 mg of zinc), as well as a blend of plant extracts from various fruits, vegetables, and herbs such as acerola, alfalfa, black currant, blueberry, elderberry, grape, grapefruit, kelp, lemon, mandarin orange, marigold, onion, orange, parsley, peppermint, rosemary, spinach, tomato, turmeric, and watercress, and quercetin granular. The placebo was formulated with inert substances such as microcrystalline cellulose, silicon dioxide, magnesium stearate, and colorants designed to match the appearance and color of the PBS.

The phytochemicals in the PBS were characterized using ultra-performance liquid chromatography–quadrupole time-of-flight mass spectrometry (UPLC-Q-TOF-MS, Waters Corp., Milford, MA, USA) equipped with a BEH C18 column (100 × 2.1 mm i.d., 1.7 µm particle size; Waters Corp.) [7]. For standardizing the identified compounds’ nomenclature within the PBS, we employed the PubChem database (accessible at https://pubchem.ncbi.nlm.nih.gov/ (accessed on 28 June 2021)), which allowed us to offer specific compound details for integration into the CODA database [8].

### 2.2. Subjects and Study Design

The clinical trial was conducted for eight weeks, utilizing a double-blinded, randomized, parallel-arm, and placebo-controlled design. Further details on the participant demographics and the study design can be found in our previous publication [5]. All subjects provided their written informed consent to analyze their metabolites and genetic material at the time of screening. The study’s protocol was approved by the Institutional Review Boards of Ewha Womans University (IRB No. 119-16) and was registered with the International Clinical Trials Registry Platform of the World Health Organization (KCT0002055).

Figure 1 illustrates the study’s comprehensive design, which included a final cohort of 84 subjects (*n* = 42/group) who completed the study, from the initial 96 enrolled subjects. Our research entailed a thorough evaluation that incorporated an analysis of 3-day dietary records, anthropometric measurements, and biochemical assessments to determine the antioxidative and anti-inflammatory effects in blood and urine samples. Furthermore, we constructed an extensive examination of metabolites in both the test materials and plasma samples. Concurrently, a target gene analysis was performed on PBMC samples collected at the beginning and end of the study, providing the essential data for our extensive systems biology analyses.

### 2.3. Biochemical Analysis

The ROS level in plasma was quantified using luminol-amplified chemiluminescence, executed on a Fluoroskan Ascent FL chemiluminescence detector (Thermo Fisher, Vantaa, Finland) [5]. The comet assay, a single-cell gel electrophoresis method, was employed to assess the DNA damage [9]. The total malondialdehyde (MDA) levels in plasma were determined using a high-performance liquid chromatography system with fluorescence detection (Shiseido, Tokyo, Japan) [10]. Additionally, the plasma levels of oxidized low-density lipoprotein, glutathione, and antioxidant enzyme activities were measured using spectrophotometry with commercially available kits (Cayman, Ann Arbor, MI, USA). Biochemical parameter differences were analyzed by employing a linear mixed-effects model (LMM), which considered random effects (subject variability), random errors (within-subject variability), and fixed effects (group, time point, and their interaction).

### 2.4. Target Gene Analysis

For the quantitative polymerase chain reaction (qPCR) array analysis, we utilized the AccuPower^®^ Customized qPCR Panel Kit (Bioneer, Daejeon, Republic of Korea) on 24 PBMC samples. Our analysis included a broad spectrum of 87 genes implicated in inflammation (47 genes), plaque formation and coagulation (3 genes), oxidation (13 genes), blood cell differentiation (2 genes), and lipid/lipoprotein metabolism (22 genes). We normalized the quantification of mRNA levels to glyceraldehyde 3-phosphate dehydrogenase (GAPDH) and calculated the relative RNA quantity using the comparative CT method. Subsequently, we pinpointed genes with a differential expression by comparing the fold changes in their relative expression between the two groups, using a threshold of >1.2-fold for upregulation and <0.8-fold for downregulation.

### 2.5. Metabolite Analysis of Plasma Samples

The metabolite analysis of plasma samples was a multistep procedure. Initially, the plasma was centrifuged after being incubated with methanol at 4 °C for one hour to precipitate the proteins. The clear supernatants were then filtered and dehydrated using a speed vacuum concentrator. A two-step derivatization process, consisting of oximation and silylation, was applied to increase the volatility of the metabolites, preparing them for the analysis. The gas chromatography–time-of-flight mass spectrometry (GC-TOF-MS) analysis utilized a comprehensive system, including an Agilent 7890 GC system (Agilent Technologies, Palo Alto, CA, USA) equipped with an Agilent 7693 autosampler and a Pegasus^®^ High-Throughput-TOF-MS system (LECO, St. Joseph, MI, USA). Chromatographic separation was performed using an Rtx-5MS column (29.8 m length × 0.25 mm i.d., 0.25 µm particle size; Restek Corp., Bellefonte, PA, USA).

Subsequently, we conducted a multivariate statistical analysis using the SIMCA software, version 14.1 (Umetrics, Umeå, Sweden). A principal component analysis (PCA) and an orthogonal partial least-squares discriminant analysis (OPLS-DA) were utilized to discern metabolite variations between and within groups. The metabolites significantly affecting group separation pre- and post-supplementation were identified by variable importance in projection (VIP) values over 0.7. Additionally, we pinpointed differential metabolites between the groups by managing the false discovery rate (FDR), setting the significance threshold at an FDR *q*-value < 0.05.

### 2.6. Identification of Biological Processes

We employed the Gene Ontology (GO) browser to identify biological processes (BPs) pertinent to the oxidative stress and inflammation phenotypes observed in our study. Initially, we pinpointed a total of 141 BPs related to oxidative stress and further organized them into 19 subcategories. These covered diverse aspects, including aging, cell redox homeostasis, responses to redox state changes, the endoplasmic reticulum stress response, DNA damage checkpoint signaling, the oxidative stress response, the ROS metabolic process, responses to nitrogen compounds, protein and lipid oxidation processes, nicotinamide adenine dinucleotide + hydrogen and nicotinamide adenine dinucleotide phosphate + hydrogen oxidation, sulfur compound metabolism, the regulation of oxidoreductase and thioredoxin peroxidase activities, heme oxidation, and host defense mechanisms. Furthermore, we identified 226 BPs linked to inflammation, which spanned a range of categories, including inflammatory responses, cytokine production, the response to cytokines, the response to prostaglandins, and the prostaglandin metabolic process.

### 2.7. Data Integration and CODA Network Analysis

We constructed a multi-layered network that incorporated phytochemicals, their targets, and BPs by integrating multi-omics data from this study, including the PBS phytochemicals, differential variables selected from clinical and omics data, and biological processes associated with oxidative stress and inflammation, all fed into the CODA database.

To investigate the complex interactions between the PBS phytochemicals and their primary targets, we employed the shortest path analysis to reveal the multi-target and overlapping effects. The prominence of the phytochemicals and targets was ascertained by their centrality within the network. Additionally, we executed Kyoto Encyclopedia of Genes and Genomes (KEGG) pathway enrichment analyses, considering pathways significantly enriched when the false-discovery-rate-adjusted *p*-value (*q*-value) was below 0.05 and the impact value exceeded 0.1. The resultant networks were visualized using the Cytoscape software, version 3.7.2, available at http://cytoscape.org/ (accessed on 2 March 2020) [11].

## 3. Results

### 3.1. Baseline Characteristics

The data analysis adhered to per-protocol principles. Table 1 details the demographic, anthropometric, and nutritional characteristics of the 84 participants in this study. The analysis revealed a close match between the placebo and PBS groups, with no significant differences noted. The subjects were healthy, middle-aged adults who typically consumed low amounts of fruit and vegetables, reflected by a recommended food score below 20 (out of a possible 47).

### 3.2. Comparative Analysis of Biomarkers: Pre- and Post-Treatment with Placebo vs. PBS

In Figure 2a, the flowchart illustrates the detailed process of merging clinical and omics data within the context of the *CODA* network analysis. Through a UPLC-Q-TOF-MS analysis, 46 compounds were identified within the PBS [5]. From these, the compounds deemed excipients essential for the formulation process were excluded, resulting in the incorporation of 33 phytochemicals into the CODA database. This exclusion accounted for 13 compounds and other entities lacking sufficient data to be definitively classified as phytochemicals (Appendix A). Moreover, to accurately depict the phenotypes of oxidative stress and inflammation, we selected 367 BPs, 141 associated with oxidative stress (Appendix A) and 226 associated with inflammatory damage (Appendix A), employing the GO browser for precision.

Figure 2b graphically displays the different biochemical and omics variables after a z-score transformation. From the 12 clinical variables assessed, significant reductions in ROS and DNA damage levels were observed in the PBS group compared to the placebo group. Additionally, the catalase activity in erythrocytes exhibited a marked increase in the PBS group. The GC-TOF-MS analysis identified 28 metabolites in plasma. Subsequently, through an OPLS-DA analysis, we identified 17 metabolites with significant differential changes between the two groups. A gene analysis of 87 candidates yielded 42 differentially expressed genes between the groups. Notably, 11 genes were upregulated in the PBS group and remained unchanged or downregulated in the placebo group (*ACACB*, *CYBA*, *ECI1*, *HMOX1*, *IFNAR1*, *NCF1*, *NFE2L2*, *TNFRSF18*, *TNFRSF1B*, *CAT*, *CYBA*, and *TLR2*). Conversely, 31 genes were downregulated in the PBS group and either remained unchanged or were upregulated in the placebo group (*CCL22*, *CXCL10*, *ESR1*, *IL12A*, *IL1R1*, *ITGAL*, *PEAR1*, *PTDSS1*, *SOCS6*, *TNF*, *TXNRD1*, *BCL6*, *CCR3*, *CD40*, *CRLF1*, *CXCL11*, *CXCL8*, *DGKQ*, *GK*, *IFNA1*, *IFNB1*, *IL12B*, *IL4R*, *IL6*, *LDLR*, *LRP1*, *MOGAT3*, *RSAD2*, *SLC27A1*, *SOCS3*, and *VCAM1*).

### 3.3. CODA Network Analyses

We utilized the CODA network platform to elucidate and visualize the complex interactions between the PBS phytochemicals, the target genes/metabolites, and the phenotypes. Our initial step involved integrating the phytochemical profiles from the PBS samples and BPs into the CODA system as foundational data. As new data were introduced, the network expanded, revealing more potential connections. This iterative process of network growth continued until we achieved a comprehensive dataset. The resulting four-layered association network, shown in Figure 3a, consisted of 5545 nodes and 39,818 edges, illustrating the potential associations among 12 phytochemicals, 4088 genes, 5 metabolites, and 222 BPs. Notably, only 12 of the 35 phytochemicals initially considered were incorporated into the network. These included quercetin, ellagic acid, wogonin, quercitrin, phloretin, chlorogenic acid, carnosic acid, hesperidin, demethoxycurcumin, isoquercitrin, curcumin, and isorhamnetin 3-o-glucoside. From the 367 BPs evaluated, 222 were integrated, with 62 related to oxidative stress and 160 to inflammation.

Subsequently, the network was refined to retain only the targets directly linked to differential omics markers. This refinement considerably streamlined the network to 255 nodes and 1579 edges, as shown in Figure 3b. This phase excluded two phytochemicals (isoquercitrin and isorhamnetin 3-o-glucoside) and four oxidative-stress-related categories (response to redox state, sulfur compound metabolic process, heme oxidation, and response to host defenses).

### 3.4. Diverse and Synergistic Interactions between Phytochemicals and Primary Targets

Several PBS phytochemicals demonstrated interactions with a multitude of targets, shedding light on their extensive range of effects. Figure 4a shows that quercetin interacted with the highest number of targets (degree = 24), followed by ellagic acid and hesperidin (degree = 6), chlorogenic acid (degree = 5), and quercitrin (degree = 4). Additionally, certain targets were linked to more than two phytochemicals, indicating possible synergistic effects when the phytochemicals engaged with specific genes. Genes associated with antioxidative functions, such as *HMOX1* (degree = 9) and *NFE2L2* (degree = 6), were upregulated and had numerous connections to phytochemicals. Conversely, genes such as *TNF* (degree = 8), *IL-6* and *CXCL8* (degree = 4), and *ESR1*, *IL1R1*, *LDLR*, *TNFRSF1B*, and *VCAM1* (degree = 2) were found to be downregulated.

### 3.5. Construction of Target-Specific Signaling Pathways

To delve into the mechanisms at play, we carried out a KEGG pathway enrichment analysis. This analysis highlighted that the PBS impacted several pathways, including the Toll-like receptor (TLR), Janus kinase/signal transducer and activator of transcription (JAK-STAT), RIG-I-like receptor (RLR), nuclear factor kappa-light-chain-enhancer of activated B cells (NF-κB), cytosolic DNA-sensing, adipocytokine, and C-type lectin receptor (CLR) signaling pathways, which are pivotal in oxidative stress and inflammation. Specifically, the analysis detailed the involvement of five principal phytochemicals: quercetin, ellagic acid, hesperidin, chlorogenic acid, and quercitrin. As illustrated in Figure 4b, quercetin is active in multiple pathways, including those regulated by the adipocytokine, CLR, and NF-kappa B signaling pathways. Meanwhile, ellagic acid, hesperidin, and chlorogenic acid are associated with the TLR signaling pathway, with both quercetin and quercitrin affecting the adipocytokine signaling pathway. These insights highlight the crucial role of these pathways in the modulation of oxidative stress and inflammation.

## 4. Discussion

The preceding randomized clinical trial (RCT) substantiated the influence of PBS supplementation on ROS scavenging and DNA damage reduction in healthy adults with a low fruit and vegetable consumption. The current study leverages a clinical data-driven network biology approach to elucidate the mechanisms by which a PBS mitigates oxidative stress and inflammation. A network analysis identified quercetin as the PBS phytochemical targeting the broadest array of genes, notably upregulating the *HMOX1* gene and downregulating the *TNF* gene. Additionally, we noted that ellagic acid, hesperidin, and chlorogenic acid jointly affected the TLR signaling pathway, while quercetin and quercitrin influenced the adipocytokine signaling pathway.

Oxidative stress and inflammation are complex physiological processes that span multiple organs, complicating their definition as single phenotypes. To navigate this complexity, we utilized BPs from the GO as descriptive tools. A BP comprises biological events enacted by one or more gene products [12]. Notably, BPs are hierarchically structured, with “child” terms specifying more detailed facets of the “parent” processes, thereby distinguishing them from pathways. For an accurate representation of oxidative stress and inflammation, we selected BPs intimately connected to these conditions and then organized them hierarchically using the GO browser. This GO analysis approach has been widely used in studies to categorize genes and annotate their roles concerning oxidative stress and inflammation [13,14,15].

Various network analysis platforms have been developed with bioinformatics, which are frequently utilized in pharmacology. These include platforms such as the Korean Traditional Knowledge Portal (http://www.koreantk.com/ (accessed on 30 November 2023), the Traditional Chinese Medicine Integrated Database [16], and KAMPO (https://kampo.ca/ (accessed on 30 November 2023)) [17]. The CODA program, however, distinguishes itself as a unique network platform. It not only captures molecular and cellular interactions, but also integrates a wealth of association data in anatomical contexts, acknowledging phenotypes influenced by diseases affecting multiple organs. Consequently, CODA leverages omics data to predict potential bioactive compound studies effectively, particularly for diseases involving unexplored functional properties or complex food constituents [6]. Utilizing CODA in conjunction with signature phytochemicals identified through UPLC-TOF-MS, we constructed a four-layered network: PBS–signature phytochemicals–target genes/metabolites–BPs, using a snowball method. This unsupervised approach allowed CODA to uncover a broad spectrum of potential health benefits provided by the PBS. This study strategically focused on oxidative stress and inflammation to elucidate the PBS’s mechanisms within a sea of vast associations, with BPs identified using the GO browser.

Over 70% of the differential markers selected in this study aligned with the CODA network. This concordance with prior findings corroborates the utility of CODA in constructing a four-layered network and supports its effectiveness in capturing the biochemical impact of the PBS. Metabolites and genes absent from the existing CODA framework may represent new potential targets for phytochemicals regarding oxidative stress and inflammation. Another notable strength of this study is its enhancement of the CODA network’s quantitative capacity. This was achieved by integrating differential markers derived from the RCT, thereby enhancing the resolution of CODA’s quantitative interpretation and enriching the analytical robustness of the study. Interestingly, of the numerous targets investigated, only 26 differential markers were regained in the refined network. Yet, the BPs affected by these markers closely reflected those identified within the more extensive network framework. These results affirm the PBS’s impact on the BPs crucial to DNA integrity and ROS metabolism, further substantiating the biological relevance of the PBS in these pathways.

Within the multi-layered framework of the CODA network, compound-target associations offer insights into the primary biological targets and potential synergistic effects of phytochemicals sourced from the PBS. The enhanced CODA network encompasses a suite of ten distinguished phytochemicals: quercetin, ellagic acid, hesperidin, chlorogenic acid, quercitrin, wogonin, phloretin, carnosic acid, demethoxycurcumin, and curcumin. Quercetin is characterized as a flavonol aglycone, while quercitrin is a glycosylated derivative thereof; hesperidin is recognized as a flavanone glycoside, and wogonin as an O-methylated flavone, all of which are classified as flavonoids [18]. The remaining phytochemicals are categorized as phenolic acids, hydrolyzable tannins, curcuminoids, and terpenoids [19]. Chlorogenic acid represents a phenolic acid, specifically a hydroxycinnamic acid. Ellagic acid is aligned with hydrolyzable tannins, a product of tannin hydrolysis. Curcumin and demethoxycurcumin are identified as curcuminoids, with the latter being a demethylated analog of curcumin, featuring two aromatic rings and O-methoxy phenolic groups. Phloretin is a member of the dihydrochalcone class, whereas carnosic acid is a phenolic diterpene. The participation of these structurally varied molecules in antioxidant and anti-inflammatory functions exemplifies the intricate nature of their structure–activity relationships.

The refined CODA network demonstrated that quercetin stood out as the most influential, showing a fourfold greater linkage with various BPs compared to the subsequent phytochemical. Known for its antioxidant and anti-inflammatory capabilities, quercetin functions as a scavenger of ROS and reactive nitrogen species, enhancing the body’s antioxidant defense [20]. It also diminishes the cytokine production induced by the lipopolysaccharide and gene expression of pro-inflammatory markers such as tumor necrosis factor alpha (TNF-α) and interleukin-8 (IL-8), through the inhibition of TLR-mediated NF-κB signaling pathways [21,22,23]. Ellagic acid contributes to lipid peroxidation inhibition, free radical neutralization, and the reduction in inflammatory cytokines and chemokines, such as TNF-α and IL-6, while suppressing NF-κB expression [24,25]. Hesperidin strengthens antioxidant defenses through the extracellular signal-regulated kinase (ERK)/nuclear factor erythroid 2-related factor 2 (Nrf2) signaling pathway [26] and obstructs the NF-κB pathway, reducing prostaglandin E2, cyclooxygenase-2, and inducible nitric oxide synthase expression [26]. Chlorogenic acid is acknowledged for its role in decreasing ROS production and the expression of pro-inflammatory mediators [27]. Quercitrin, although fifth in rank, predominantly affects oxidative-stress-related BPs, exhibiting notable anti-inflammatory and ROS-scavenging effects that include inhibiting NF-κB translocation [28,29].

The predominant targets influenced by the PBS phytochemicals include *HMOX1*, *TNF*, *IL6*, *NFE2L2*, and *CXCL8*, with *HMOX1*, the gene for heme oxygenase 1 (HO-1), being the most interconnected. HO-1 plays a crucial role in heme breakdown and provides cellular defense against oxidative and inflammatory stress by upregulating the expression of antioxidant genes [30,31,32]. Within the CODA network, *NFE2L2* is particularly associated with inflammatory BPs. This gene encodes Nrf2, a transcription factor that activates the transcription of stress response genes through the antioxidant response element (ARE) [33], thereby regulating genes involved in cellular protection, including HO-1 [34,35]. Dietary phytochemicals such as quercetin and curcumin are known to activate the *HMOX1* gene via Nrf2 modulation, offering protection against oxidative-stress-related disorders [36,37,38]. Hence, the PBS’s upregulation of *HMOX1* and *NFE2L2* could amplify its antioxidative and anti-inflammatory properties under stress conditions. Additionally, the PBS suppressed the expression of the *TNF* and *IL6* genes, which produced cytokines integral to inflammatory processes—TNF influences cell functions such as proliferation and apoptosis, while IL-6 triggers acute inflammatory responses and is implicated in chronic inflammatory diseases [39,40,41]. Clinical interventions often target TNF and IL-6 to alleviate inflammatory conditions [42,43]. These findings indicate that the diverse phytochemicals in the PBS synergistically and collectively modulate targets involved in both oxidative and inflammatory pathways, suggesting a synergistic effect.

The pathway enrichment analysis revealed that the PBS influenced crucial pathways such as the TLR, JAK-STAT, RLR, NF-κB, cytosolic DNA-sensing, adipocytokine, and CLR signaling pathways. These pathways are integral to oxidative stress and inflammation responses. Significantly, the TLR pathway, which is linked to ten genes affected by ellagic acid, hesperidin, and chlorogenic acid, includes the downregulation of *TNF*, *IFNA1*, *INFB1*, *IL6*, *IL12B*, *CXCL8*, *CXCL10*, and *CD40* and the upregulation of *TLR2* and *INFAR1*. These genes are also involved in the RLR, cytosolic DNA-sensing, and CLR pathways, crucial elements of the innate immune system. They act as pattern recognition receptors (PRRs) that detect pathogenic microorganisms and signals from cellular damage, which may initiate ROS production and provoke the release of pro-inflammatory cytokines [44,45]. Studies have consistently demonstrated that the antioxidant and anti-inflammatory actions of dietary phytochemicals target the inhibition of PRR pathways [46,47]. Thus, our findings support the hypothesis that a PBS delivers its health benefits by modulating PRR signaling pathways.

In this research, we discovered that quercetin affects additional signaling pathways, including those of NF-κB, CLR, and adipocytokine. Specifically, quercetin influences crucial genes within these pathways: *TNF*, *CXCL8*, *CD40*, *VCAM1*, and *IL1R1* in the NF-κB pathway; *IL6*, *TNF*, and *IL12B* in the CLR pathway; and *TNF*, *TNFRSF1B*, and *SOCS4* in the adipocytokine pathway. The NF-κB pathway, central to oxidative stress and inflammation, orchestrates the transcription of genes linked with pro-inflammatory cytokines, adhesion molecules, antioxidant enzymes, and proteins integral to the acute inflammatory response [48]. Typically inactive under resting conditions, NF-κB is sequestered in the cytoplasm, bound to the IκB inhibitor. When activated by ROS or pro-inflammatory stimuli, IκB is phosphorylated and degraded, releasing NF-κB to translocate to the nucleus and initiate gene transcription [49]. Additionally, CLRs, a varied family of carbohydrate-binding proteins, are essential for the immune system, recognizing a spectrum of ligands, including those involved in pathogen-associated and damage-associated molecular patterns. Lastly, the adipocytokine signaling pathway, which involves adipokines such as leptin and adiponectin, plays essential roles in metabolic regulation, immune responses, and the mediation of inflammation and oxidative stress [50]. Quercitrin, formed from the conjugation of quercetin and the sugar rhamnose, is known for its antioxidant capabilities, which include the neutralization of free radicals [51]. Our findings also indicate quercitrin’s role in modulating the adipocytokine signaling pathway. Thus, the interaction with these pathways may explain the beneficial effects of the PBS in mitigating oxidative stress and inflammation.

The limitations of this study warrant careful acknowledgment. Primarily, the PBS contained a range of vitamins and minerals at doses corresponding to 50–100% of the recommended dietary reference intake. Crucially, certain micronutrients within this profile, such as vitamin C, vitamin E, and selenium, are recognized for their antioxidant capabilities, which likely significantly influenced the PBS’s antioxidative potential. Additionally, the CODA framework was challenged by its inability to accurately discern specific entities in the PBS, including various isoforms with glycoside moieties, restricting their subsequent analytical consideration. Since phytochemicals occur in many secondary metabolite configurations in plants, issues with bioavailability and bioaccessibility remain prevalent. LC-MS was chosen as the analytical method to select compounds in this study due to the inherent difficulties in ascertaining their bioactive states. Future investigations are recommended to examine these secondary metabolites and their aglycone counterparts as prospective bioactive constituents. Furthermore, the application of qPCR for a transcriptomic analysis in this study confined the validation of numerous genes postulated by CODA. Employing microarray technology may furnish a more comprehensive scope for transcriptomic exploration, enhancing the genomic coverage and substantiating the identification of target genes. Finally, it must be noted that the outcomes derived from in silico models depend on the extensiveness of the existing literature and the database integrity. As a result, network analyses are susceptible to inherent biases tied to the quantity of the gathered empirical evidence.

## 5. Conclusions

This study utilized a clinical data-driven network system methodology, providing new insights into the complex and synergistic actions of phytochemicals in a PBS and their potential mechanisms. We identified signature phytochemicals such as quercetin, ellagic acid, hesperidin, chlorogenic acid, and quercitrin that target an array of pathways, including the Toll-like receptor, NF-kappa B, adipocytokine, and C-type lectin receptor signaling pathways. These phytochemicals influence genes such as *HMOX1*, *TNF*, *NFE2L2*, *CXCL8*, and *IL6*. Nonetheless, this investigation is subject to certain limitations that warrant additional inquiries. Notably, the contributions of vitamins and minerals present in the PBS to the observed phenotypes were not methodically evaluated, and the post-absorption metabolism of the phytochemicals remained unexplored. Furthermore, gene expression profiling was performed using qPCR, a technique that, while precise, does not provide the comprehensive genomic coverage achieved using a microarray analysis. Additionally, the predictive outcomes of the in silico models were constructed upon the foundation of pre-existing literature and databases, which could potentially overlook emergent molecular interactions. Despite these challenges, this systems biology approach has substantially enhanced our comprehension of the PBS’s effects by mapping out the complex interplay between multiple phytochemicals and their targets, suggesting possible molecular mechanisms.

## Figures and Tables

**Figure 1 antioxidants-13-00036-f001:**
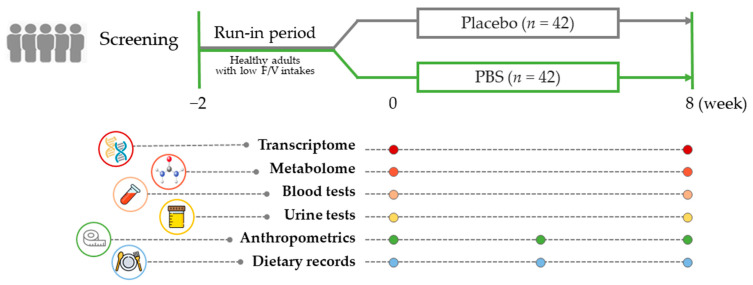
Study design summary: After a 2-week run-in period, adults with a low fruit and vegetable intake were randomly assigned to receive either a placebo or the PBS for 8 weeks. Blood and urine samples were collected while fasting at the beginning and end of the study to evaluate clinical and multi-omics biomarkers. PBS, a plant-based vitamin/mineral supplement.

**Figure 2 antioxidants-13-00036-f002:**
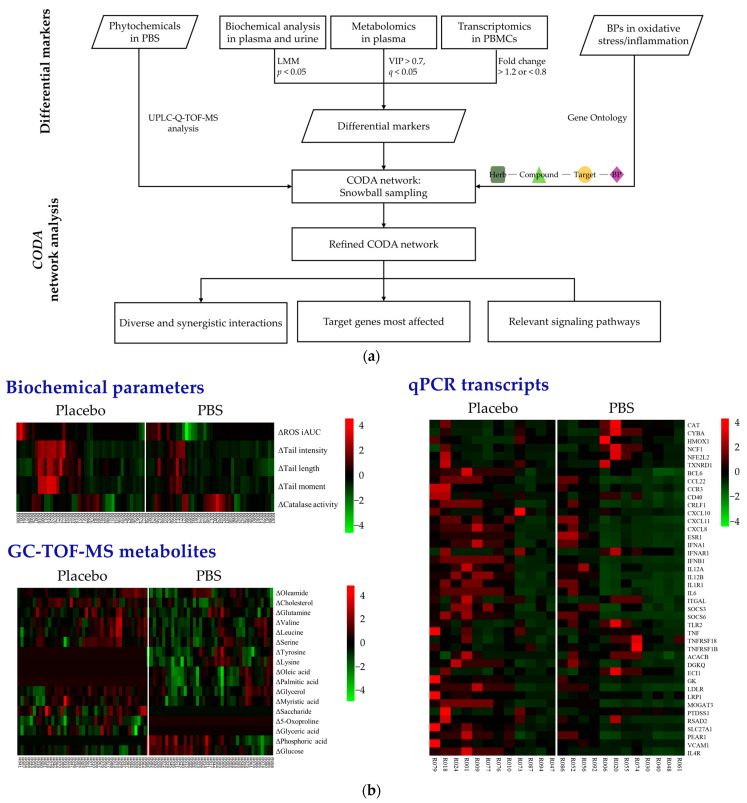
Integration and analysis of clinical and multi-omics data through *CODA* network analysis: (**a**) Comprehensive workflow. (**b**) Heatmaps indicate significant changes after 8 weeks of PBS supplementation. A color gradient was used, where red represents increased levels, green denotes reduced levels, and black indicates no significant change. PBS, a plant-based vitamin/mineral supplement; BP, biological process; LMM, linear mixed-effects model; VIP, variable importance in projection; PBMC, peripheral blood mononuclear cell; CODA, context-oriented directed association; GC-TOF-MS, gas chromatography–time-of-flight mass spectrometry; qPCR, quantitative polymerase chain reaction.

**Figure 3 antioxidants-13-00036-f003:**
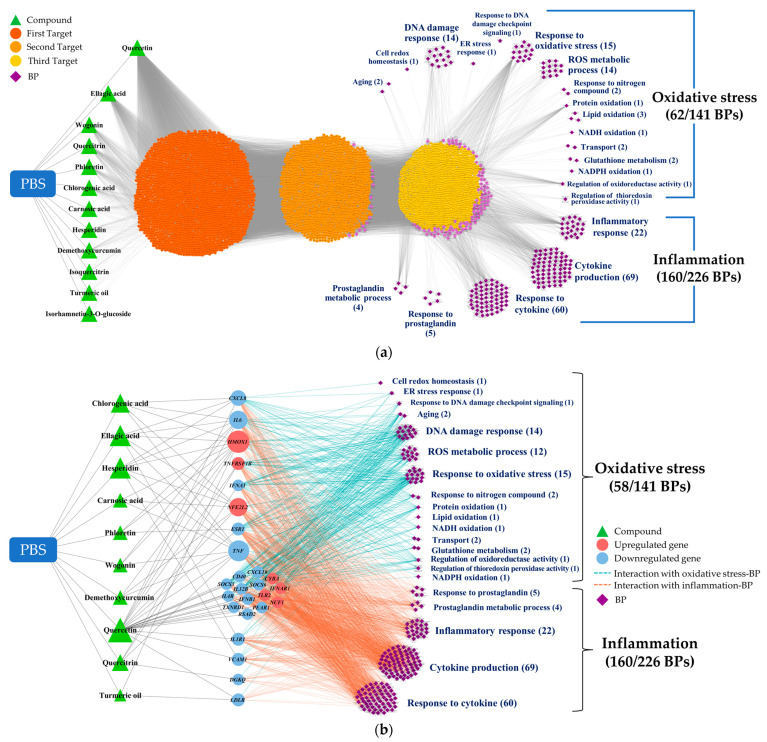
*CODA* network visualization for the complex interplay of components within the study: (**a**) A four-layered network, delineating possible linkages between PBS phytochemicals, their targets, and BPs, as inferred through snowball sampling methodology. (**b**) A streamlined network, emphasizing targets directly linked to altered omics markers. CODA, context-oriented directed association; PBS, a plant-based vitamin/mineral supplement; BP, biological process; ER, endoplasmic reticulum; NADH, nicotinamide adenine dinucleotide + hydrogen; NADPH, nicotinamide adenine dinucleotide phosphate + hydrogen.

**Figure 4 antioxidants-13-00036-f004:**
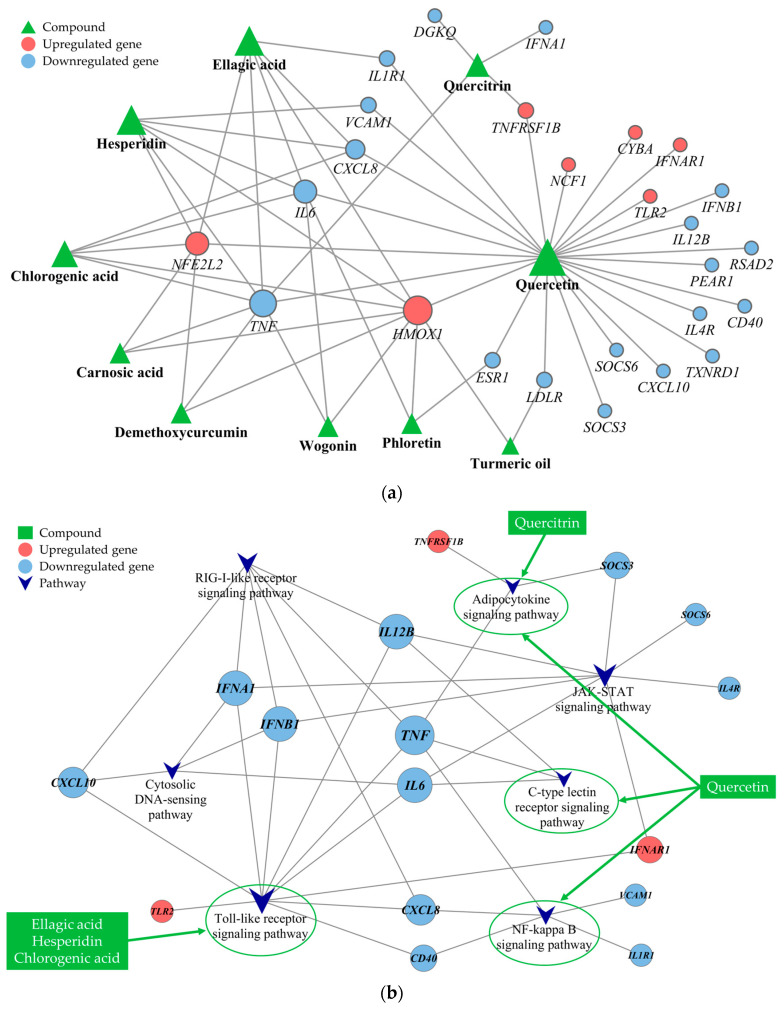
Phytochemical interactions and pathways: (**a**) the complex and potentially synergistic interactions among various phytochemicals and (**b**) specific signaling pathways influenced by these phytochemicals. The target-pathway network constructed gives precedence to targets associated with phytochemical compounds and significant pathways, characterized by a *q*-value < 0.05 and an impact > 0.1. The node size in the network is proportional to the node’s degree of connectivity.

**Table 1 antioxidants-13-00036-t001:** Baseline characteristics of participants in the per-protocol analysis.

Variables	Placebo (*n* = 42)	PBS (*n* = 42)	*p*-Value
Age (year)	41.6 ± 1.7	38.2 ± 1.7	0.169
Sex (men/women, *n*)	13/29	13/29	1.000
Recommended food score	19.5 ± 1.3	19.1 ± 1.5	0.830
Body weight (kg)	67.4 ± 2.1	65.1 ± 2.2	0.462
Body mass index (kg/m^2^)	24.8 ± 0.6	23.7 ± 0.6	0.202
Percent of body fat (%)	31.7 ± 0.9	30.1 ± 1.0	0.258
Smoker, *n* (%)	3 (7.1)	4 (9.5)	0.693
Alcohol drinker, *n* (%)	22 (52.4)	24 (57.1)	0.661
Vital signs			
Pulse rate (beats/min)	81.9 ± 1.6	85 ± 1.7	0.199
Temperature (°C)	35.7 ± 0.1	35.6 ± 0.1	0.336
Blood pressure (mmHg)			
Systolic blood pressure	119.1 ± 2.1	116.7 ± 2.0	0.414
Diastolic blood pressure	79.5 ± 1.6	79.0 ± 1.5	0.786

Continuous variables are presented as mean ± standard error, and categorical variables are shown as frequencies. Student’s *t*-test was employed to assess differences in continuous variables between the placebo and PBS groups, while the chi-squared test was used for categorical variables. PBS, a plant-based vitamin/mineral supplement.

## Data Availability

The data are available from the authors upon request.

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
