# Peer review of "Diverse and Synergistic Actions of Phytochemicals in a Plant-Based Multivitamin/Mineral Supplement against Oxidative Stress and Inflammation in Healthy Individuals: A Systems Biology Approach Based on a Randomized Clinical Trial"

_antioxidants, 2023, doi:10.3390/antiox13010036_

Round 1
Reviewer 1 Report
Comments and Suggestions for Authors
The authors produced an interesting paper describing the antioxidant and anti-inflammatory effect of a product, namely PBS, on healthy people with low daily intake of vegetables. The experimental design is well established and useful for reaching the aims of the work. Furthermore, the authors used an innovative and interesting approach integrating in silico methods in their study. Overall, the article is well written, and authors correctly specified the limitations of their study, showing great awareness of their work. For all this reasons, I recommend the article to be published.
There are just two points that I would like to bring to the attention of the authors:
· To increase the quality of their work, it is necessary to explain the rationale behind the association of all the components of the product (PBS) they used.
· The quality of the images needs to be improved.
Reviewer 2 Report
Comments and Suggestions for Authors
The manuscript entitled “Diverse and synergistic actions of phytochemicals in a plant-based multivitamine/mineral supplement against oxidative stress and inflammation in healthy individuals: A system biology approach based on a randomized clinical trial” by Seunghee Kang et al. tried to combine the phytochemicals with biological phenomena in human bodied.
This trial is quite challenging and interesting one, however, the authors have to be very careful to carry out experiments considering the ethical laws and rules.
(Permission 0f the government (university) must be necessary)
In general
1, The trial was successful enough to provide new observations, although the reason for the new observation is still unclear. The reason for this comes from the lacking the structure -activity relationship.
For example, the authors picked out five phytochemicals, quercetin, quercietrin, ellagic acid, hesperidin, and chlorogenic acid. Three compounds are glycosylated and two compounds are aglycone. Three compounds are flavonoid derivatives, two compounds are polyphenol derivatives.
The authors did not argue these points in the manuscript.
2, The structure of the compound is important, it is true, in addition the quantity of the compound is also important in the biological phenomena observed. The authors provide some results concerning the quantitative analysis of the PBS.
3, In connection with the previous point, the amounts of minerals and vitamins also contribute the biological phenomena observed. In this meaning, it is also highly desirable to provide some evidences for this point.
The author’s trial is meaningful enough to be acceptable for publication in Antioxidants, however, it is necessary to add some more discussion and explanations in the manuscript
Comments on the Quality of English Language
The English usage is not so bad. For the brushing up, it is better to help by the support (help) of native English writer or editorial office of the journal.
Comments to the editor
